# Seasonal Variation of Glucosinolate Hydrolysis Products in Commercial White and Red Cabbages (*Brassica oleracea* var. *capitata*)

**DOI:** 10.3390/foods9111682

**Published:** 2020-11-17

**Authors:** Nicole S. Wermter, Sascha Rohn, Franziska S. Hanschen

**Affiliations:** 1Plant Quality and Food Security, Leibniz-Institute of Vegetable and Ornamental Crops (IGZ), Theodor-Echtermeyer-Weg 1, 14979 Grossbeeren, Germany; nicolewermter@yahoo.de; 2Hamburg School of Food Science, Institute of Food Chemistry, University of Hamburg, Grindelallee 117, 20146 Hamburg, Germany; rohn@chemie.uni-hamburg.de; 3Department of Food Chemistry and Analysis, Institute of Food Technology and Food Chemistry, Technische Universität Berlin, TIB 4/3-1, Gustav-Meyer-Allee 25, 13355 Berlin, Germany

**Keywords:** glucosinolate, cabbage, isothiocyanate, epithionitrile, nitrile, *Brassica*, seasonal variation, food retailer

## Abstract

*Brassica* vegetables contain glucosinolates, which are well-known for their potential to form health-promoting isothiocyanates. Among those crucifers, white and red cabbage are commonly consumed vegetables, exhibiting different glucosinolate and hydrolysis profiles thereof. Regarding the health beneficial effects from these vegetables, more information, especially concerning the seasonal variation of glucosinolate profiles and the formation of their bioactive hydrolysis products in commercial cabbages, is needed. In this study, glucosinolates and glucosinolate hydrolysis product profiles in red and white cabbages from three different food retailers were monitored over six different sampling dates across the selling season in autumn. For the first time, it was shown that, while glucosinolate profiles were similar in each cabbage variety, glucosinolate hydrolysis product profiles and hydrolysis behavior varied considerably over the season. The highest total isothiocyanate concentrations were observed in conventional red (1.66 μmol/g FW) and organic white (0.93 μmol/g FW) cabbages purchased at the first sampling date in September. Here, red cabbage was with up to 1.06 μmol/g FW of 4-(methylsulfinyl)butyl isothiocyanate (sulforaphane), an excellent source for this health-promoting isothiocyanate. Cabbages purchased 11 weeks later in autumn released lower levels of isothiocyanates, but mainly nitriles and epithionitriles. The results indicate that commercial cabbages purchased in early autumn could be healthier options than those purchased later in the year.

## 1. Introduction

With a consumption of 5.2 kg/person and year in 2017/2018, red cabbage (*Brassica oleracea* var. *capitata* f. *rubra*) and white cabbage (*Brassica oleracea* var. *capitata* f. *alba*) are the most consumed Brassicaceae vegetables in Germany, contributing to 5% of total vegetable intake [1]. These vegetables are rich in glucosinolates (GLSs), secondary, sulfurous plant constituents, which are particularly present in vacuoles of plant cells of the Brassicaceae family. The chemical GLS structure is determined by the nature of the side chain, depending on the amino acid inserted during biosynthesis [2,3] and GLSs typically contain a glucose unit, bound with a central carbon atom with nitrogen grouping via a thioether bridge. The carbon atom, in turn, is linked to a sulfate group and an organic aglycone residue, possessing an alkyl, alkenyl, aryl, or indole group [2]. This organic residue is inherent to the GLS, its chemical properties and its flavor, respectively [4,5]. Upon attack by herbivores or due to cutting or chopping of vegetables rich in GLSs, GLSs, which were previously present in spatially separated cell vacuoles, are hydrolyzed by myrosinase to various herbivore-toxic degradation products [6,7]. Hydrolysis by myrosinase occurs due to enzymatic cleavage of the thioglycoside bond, first resulting in an unstable aglucone (thiohydroximate-*O*-sulfate). The aglucone can then undergo a lossen-like rearrangement to form isothiocyanates (ITCs) or decompose into nitriles and molecular sulfur. Moreover, in the presence of certain proteins such as the epithiospecifier protein (ESP), an aglycone with a terminal double bond favors epithionitrile (ETN) formation [7].

Consumption of ITCs can positively affect human health as they have antimicrobial, antidiabetogenic, chemopreventive, and anticarcinogenic properties [8,9]. Previous studies have shown a positive correlation between ITC uptake and cancer prevention [8,10,11], and especially 4-(methylsulfinyl)butyl ITC (sulforaphane; 4MSOB-ITC) is valued for its anticarcinogenic potential [8]. Simple nitriles and ETNs on the other hand, seem to have less health beneficial effects [12,13]. Studies have elucidated that *Brassica* vegetables, not only ITCs, but also nitriles and ETNs can be the most predominant degradation products [14,15]. Consequently, in order to estimate the health beneficial potential of *Brassica* vegetables, it is of great importance to not only analyze intact GLSs, but also their behavior during hydrolysis.

The natural GLS content in vegetables varies in accordance with genotype, plant developmental step, soil and cultivation conditions, and other ecophysiological influences, but it is also affected by storage [14,16,17]. Moreover, GLS levels vary over the growth season and several studies reported higher GLS levels in *B. oleracea* plants grown in spring compared to plants grown in autumn [17], while in broccoli (*B. oleracea* var. *italica*), GLS levels were higher when grown in the summer season compared to the spring season [18]. Similarly, Nuñez-Gómez et al. (2020) recently reported higher GLS levels in broccoli grown in autumn compared to broccoli grown in the spring season. Moreover, when comparing two spring seasons, the one with less rainy days and higher temperature resulted in higher GLS levels [19]. Experiments performed under controlled conditions indicate that temperature, as well as day length affect GLS biosynthesis in *B. oleracea* in a structure-dependent way [20,21,22].

In order to predict GLS-based health beneficial properties, more information especially concerning GLS profiles and especially on their hydrolysis behavior in *Brassica* foods available for the consumer, is needed. Possessing a long harvesting period—typically ranging from June to November—red and white cabbages are facing seasonal changes and different ecophysiological influences, which can have a great impact on the GLS profile [16,20] and might also affect the potential to form health-preventive ITC. To date, little is known about how GLS hydrolysis products are affected by different cultivation conditions. No revealing insight has been given on how GLS, and even more importantly, the formation behavior of their hydrolysis products varies in commercial cabbages across the whole season. Therefore, the objective of this study was to monitor the variation of GLS levels and the formation of their bioactive hydrolysis products in commercial red and white cabbage heads obtained from three local retailers in Germany and to link the data with the cultivation practices and post-harvest storage conditions.

## 2. Materials and Methods

### 2.1. Chemicals

Allyl GLS (Allyl; ≥99%, reference compound), 4-hydroxybenzyl GLS (4OHbenzyl; purity ≥99%, internal standard), and methylene chloride (GC Ultra Grade, solvent) were purchased from Carl Roth GmbH and Co. KG (Karlsruhe, Germany). Allyl ITC (Allyl-ITC; ≥99%, reference compound), benzonitrile (≥99.9%, internal standard), DEAE-Sephadex A-25 (anion exchanger), and the reference compounds 3-butenenitrile (Allyl-CN; ≥98%), 4-pentenenitrile (3But-CN; ≥97%), 3-phenylpropanenitrile (≥99%), were obtained from Sigma-Aldrich Chemie GmbH (Steinheim, Germany). The reference compounds 3-butenyl ITC (3But-ITC; ≥95%) and 4-pentenyl ITC (≥95%) were purchased from TCI Deutschland GmbH (Eschborn, Germany). 3-(Methylsulfinyl)propyl ITC (3MSOP-ITC) and 4-(methylsulfanyl)butyl ITC (4MTB-ITC; ≥98%) were purchased from Santa Cruz Biotechnology (Heidelberg, Germany). 4-(Methylsulfinyl)butyl ITC (4MSOB-ITC) was purchased from Enzo Life Sciences GmbH (Lörrach, Germany). The ETN 1-cyano-2,3-epithiopropane (CETP; ≥95%) was synthesized by Taros Chemicals GmbH and Co. KG (Dortmund, Germany) and 1-cyano-3,4-epithiobutane (CETB; ≥95%) was synthetized by ASCA GmbH Angewandte Synthesechemie Adlershof (Berlin, Germany). 5-(Methylsulfanyl)pentanenitrile (4MTB-CN) and 5-(methylsulfinyl)pentanenitrile (4MSOB-CN) were purchased from Enamine (SIA Enamine, Latvia, Riga). 3-Butenyl GLS (3But; ≥95%), 2(*R*)-hydroxy-3-butenyl GLS (2OH3But; ≥98%), 4-(methylsulfanyl)butyl GLS (4MTB; ≥98%), 4-(methylsulfinyl)butyl GLS (4MSOB; ≥98%), 3-(methylsulfinyl)propyl GLS (3MSOP; ≥98%), and 2-phenylethyl GLS (2PE; ≥98%) were acquired from Phytolab GmbH and Co. KG, Vestenbergsgreuth, Germany. The solvents methanol (≥99.95%), acetonitrile (LC-MS grade), and arylsulfatase (enzyme) were purchased from Th. Geyer GmbH and Co. KG (Renningen, Germany).

### 2.2. Plant Material

Three red cabbages (*Brassica oleracea* convar. *capitata* var. *rubra* L.) and three white cabbages (*Brassica oleracea* convar. *capitata* var. *alba)* were purchased at regular intervals from the same two local conventional supermarkets (CON_1_, CON_2_) and from the same organic supermarket (ORG_1_) in Brandenburg, Germany over a period of 3 months (September–November 2019). The supermarkets were selected with regard to different German food trading companies. The two conventional supermarkets selected belong to the two biggest food trading companies in Germany and the organic supermarket also belongs to a big organic food supermarket chain. The exact sampling dates (S1–S6) are given in Table 1. Additionally, red and white cabbage heads, grown in the field at the Leibniz-Institute of Vegetable and Ornamental Crops (IGZ) in Grossbeeren, Germany, were harvested freshly in order to compare GLSs and their hydrolysis products with commercial cabbages. Therefore, red (cultivar ‘Redma RZ F1′) and white (cultivar ‘Dottenfelder Dauer’) cabbage seeds were sown (13.06.2019 and 20.06.2019) on loamy soil (pH 7.3) and then grown for 3 months at the IGZ (52°20′59.0″ N 13°18′57.5″ E). The red cabbage was cultivated with 100% of the required nitrogen level and fertilized using calcium ammonium nitrate and Patentkali^®^ (419 kg N/ha). Before cultivation, the field was fertilized once with calcium ammonium nitrate (CAN) and Patentkali^®^ and later fertilized a second time, with CAN only during the cultivation period. In total, 377.78 kg CAN/ha and 40.74 kg Patentkali^®^ were applied for fertilization.

For white cabbage, Aminofert^®^ Vinasse fertilizer (BayWa AG, Munich, Germany) was applied (60 kg N/ha) with 30% of the required nitrogen level. Here, fertilization occurred before the sowing of the seeds on 06 June 2019 and a second time on 25 June 2019 for head formation. The origin and cultivation background of commercial cabbages were investigated by interrogating service staff at supermarkets and by contacting growers. Cabbages sold at CON_1_ and ORG_1_ were procured from different farming areas in northern Germany (CON_1_: Neuenkirchen and Helse (Schleswig-Holstein), Germany; ORG_1_: Blankensee (Mecklenburg-Western Pomerania), Hedwigenkoog (Schleswig-Holstein), Vierlinden and Seeblick (Brandenburg, Germany), whereas cabbages purchased from CON_2_ could continuously be procured from the same farming area, but on different fields within a 30 km perimeter in Neuenkirchen, Schleswig-Holstein, Germany) (Table 2). With regard to the cultivars, mainly white cabbage varieties such as “Storema”, “Lennox”, “Marcello”, and “Impala” and red cabbage varieties, especially “Futurima”, “Rodima”, “Bandolero”, and “Klimaro” were cultivated in the Dithmarschen region (Helse, Neuenkirchen) for CON_1_ and CON_2_. Cultivars “Rodynda”(red cabbage) and “Dowinda” (white cabbage) were mainly cultivated in Hohennauen, Germany for ORG_1_. Early cabbage cultivars (“Marcello”, “Bandolero”) were likely purchased between 04 and 05 September 2019, whereas later cabbage cultivars (“Storema”, “Lennox”, “Impala”, “Futurima”, “Klimaro”, “Rodynda”, “Dowinda”) could be purchased between 09 September 2019 and 06 November 2019. 

Long-term stored white cabbages (‘Storema”, “Lennox”, “Impala”, “Dowinda”) were purchased as of 18 November 2019 and stored at 0.1–0.3 °C in warehouses at the wholesaler. Cabbages from CON_1_ and CON_2_ were fertilized using a combination of urea, calcium ammonium nitrate, phosphate and potassium, whilst organically cultivated cabbage ORG_1_ was fertilized using *hair-meal pellets* (200 kg N/ha) or compost (Table 2). According to growers in Blankensee, Neuenkirchen, Vierlinden, and Helse, only hybrid cultivars such as “Storema”, “Impala”, “Lennox”, and “Bandolero” were grown, harvested, and later sold as ripe cabbages to CON_1_ and CON_2_, whereas non-hybrid cultivars such as “Rodynda” and “Dowinda” (grown for ORG_1_) were generally grown in Hedwigenkoog and Hohennauen. According to growers in Helse, seeds were sown from the 16th to the 20th calendar week of 2019, and cabbages were harvested from the 23rd to the 46th calendar week on heavy, sea marsh soil. Similar sowing and harvesting dates also applied for cabbages cultivated in Neuenkirchen, which were harvested between weeks 23 to 45 and also grown on heavy, sea marsh soil. According to growers in Helse, which supplied CON_1_ with cabbage, cabbage heads grown for the conventional market generally grew slower in the Dithmarschen region and were grown for 150 d in Helse on sea marsh soil (late cabbage). The growth of early cultivars (S1 sampling of CON_1_ and CON_2_) was accelerated with non-woven fibre barriers. The alleged storage conditions, according to all growers before the selling period and conditions in the supermarket during selling time, according to salespersons in CON_1_, CON_2_, and ORG_1_ are listed in Table 2, while in Appendix A all information collected during this study on the analyzed cabbages are given for each cabbage separately and in more detail.

In order to give a better understanding of how GLSs and their hydrolysis products can differ in commercial cabbages and how they might change within the season, GLSs and their hydrolysis products were additionally monitored by comparing cabbage heads from an organic and two conventional supermarkets (ORG_1_, CON_1_, CON_2_) with cabbage heads grown at the Leibniz-Institute of Vegetable and Ornamental Crops (IGZ), Grossbeeren, Germany. A further aim of this work was to link the results with the common cultivation practices and the alleged storage conditions (Table 2).

### 2.3. Sample Preparation

Fresh cabbage heads were chopped in half. Of the two obtained halves, one of the halves was halved again, and two quarters were obtained. Afterwards, one of the obtained quarters was divided into 2–3 strips (weight: 70–180 g, width: 1–1.5 cm) along the middle, and the strips were frozen at −20 °C overnight before lyophilization (11 d) and were later ground. The remaining plant material of the same quarter, from which the strips were obtained, was then cut into small pieces of 1 cm width. The chopped plant material was thoroughly mixed by hand, and 15–20 g fresh cabbage was then given into a round bottom glass vessel for homogenization. Afterwards, 15–20 mL of distilled water was added, in order to obtain a 1:1 ratio of plant material and water. Then, samples were homogenized for 1 min at a rate of 20,000 rpm using a mixer (H04, Edmund Bühler GmbH, Tübingen, Germany) and incubated for 1 h at room temperature (22 °C).

### 2.4. Analysis of Glucosinolates

To determine the profiles and concentrations of GLS in red and white cabbages, 10 mg of lyophilized powder was extracted and GLS was analyzed as their desulfo-form [23]. Briefly, 10 mg of dry plant powder was extracted thrice using 70% of hot methanol in the presence of 0.025 μmol 4-hydroxybenzyl GLS as an internal standard. The extracts were combined and desulfated on a DEAE-Sephadex A-25 ion-exchanger column using aryl sulfatase. Afterwards, desulfo-GLSs were eluted with 1 mL of water and analyzed using an Agilent UHPLC-DAD-ToF-MS system equipped with a Poroshell 120 EC-C18 column (100 × 2.1 mm, 2.7 μm; Agilent Technologies), a gradient of water, and 40% acetonitrile, as described previously [23]. Desulfo-GLSs were quantified at 229 nm via the internal standard and the calibration factor reported in the DIN EN ISO 9167-1 and calculated on this basis for 4-hydroxybenzyl GLS.

### 2.5. Determination of Glucosinolate Breakdown Products

For the analysis of GLS hydrolysis products released from red and white cabbage tissue, the protocol described by Hanschen and Schreiner (2017) was followed with small modifications [14]: Briefly, 500 mg of the homogenized fresh samples (containing 50% of water) were weighed into solvent resistant centrifuge tubes. During the first two samplings, 1 g of sample homogenate was used, which might have led to reduced recoveries for nitriles and epithionitriles, due to higher water to solvent ratio. Then, the internal standard benzonitrile (0.2 µmol) was added and GLS hydrolysis products were extracted 3 times using methylene chloride: 2 mL during the first extraction and 1.5 mL of methylene chloride in the second and third extraction. Then, samples were analyzed as described previously [14], except that in the present study an Agilent J&W VF-5ms GC-MS column (30 m × 0.25 mm × 0.25 µm) coupled to a 10 m EZ-Guard P/N:CP9013 column was used for analyte separation.

### 2.6. Statistical Analysis

To investigate differences between different sampling dates (S1–S6), means were compared using ANOVA and Tukey’s HSD test and STATISTICA version 13.5.0.17 software (TIBCO Software Inc., Palo Alto, CA, USA) with a significance level of *p* ≤ 0.05. All analyses were carried out in triplicate by analyzing three biological replicates.

## 3. Results

Samples (three cabbage heads) from each of the three retailers (conventional supermarkets CON_1_, CON_2_, and organic supermarket ORG_1_) were collected between September and November 2019 every 2 to 3 weeks, summing up to a total of six sampling dates (S1–S6). GLS-profiles and the corresponding GLS hydrolysis products were monitored over the six sampling periods and additionally compared to fresh samples (four red and white cabbage heads) harvested from a field at the IGZ in Grossbeeren, Germany. Most analyzed cabbage heads differed in their regional origin and harvest time (Table 2). The dates of the purchased or harvested cabbages over the six sampling periods (S1–S6) are listed in Table 1.

### 3.1. Glucosinolates in White and Red Cabbage from Local Food Retailers

The GLS profile of the most abundant GLS of white cabbage purchased from the different food retailers over the 3-month period is given in Figure 1A–C, whilst the GLS profile for red cabbage is displayed in Figure 1D–F. Table 3 shows the chemical structures of the most abundant cabbage GLS, as well as their GLS hydrolysis product names including the abbreviations. In the heads of the analyzed white and red cabbage cultivars, a total of 12 chemically different GLSs were detected (Appendix A). The main GLSs were allyl GLS (Allyl), 3-butenyl GLS (3But), 2-hydroxy-3-butenyl GLS (2OH3But), 3-(methylsulfinyl)propyl GLS (3MSOP), 4-(methylsulfinyl)butyl GLS (4MSOB), and indol-3-ylmethyl GLS (I3M) (Figure 1). The GLS profile in white and red cabbage cultivars between different supermarkets was often found to be similar within the same cabbage variety. In that way, Allyl, 3MSOP, and I3M were found to be most dominant in white cabbage, with maximum Allyl concentrations reaching 0.71 ± 0.02 μmol/g FW in S5 (Figure 1A), up to 0.64 ± 0.17 μmol/g FW 3MSOP in S1 of ORG_1_, and up to 0.49 ± 0.12 µmol/g FW I3M in S4 of ORG_1_ (Figure 1C). Red cabbage was often the richest in 2OH3But and 4MSOB (Figure 1D–F), with up to 0.99 ± 0.19 μmol/g FW 2OH3But in S1 of CON_1_ (Figure 1D) and 0.93 ± 0.05 μmol/g FW 4MSOB in S1 of CON_1_ (Figure 1D), respectively. However, Allyl, 3But, 3MSOP, and I3M were also formed in considerable amounts in red cabbage (Figure 1D–F).

In general, red cabbage heads produced higher levels of GLSs (Figure 1D–F) compared to white cabbage heads (Figure 1A–C). The highest total GLS levels in red cabbage heads were detected in S1 from CON_1_ (3.12 ± 0.27 μmol/g FW) (Figure 1D), whereas the highest total GLS content for white cabbage was observed in S5 from CON_1_ (1.61 ± 0.11 μmol/g FW) (Figure 1A).

Whilst general increases in total GLS concentration were found from sampling S1 to S6 (especially due to Allyl) in white cabbages from CON_2_ (Figure 1B), a general decreasing trend in total GLS concentration was detected in white cabbages procured from ORG_1_ (Figure 1C). In red cabbage, total GLSs varied for the individual samples from the same food retailers in all purchased red cabbage heads (Figure 1D–F), and no specific trend was noted between purchase dates from the different retailers. With regard to individual GLSs, many GLSs in cabbages from the three supermarkets did not significantly change over the different sampling dates, such as GLSs in red cabbage from CON_2_ (Figure 1E), while others were affected (Figure 1).

The two major GLSs detected in white cabbage heads were 3MSOP and Allyl. In white cabbages procured from CON_1_ (S1-S6, Figure 1A), Allyl increased by 2.1-fold from S1 to S5 and then decreased in S6 to levels similar to S1. In white cabbages, which were purchased from CON_2_, a general increase in Allyl content was observed from S1 to S6 (0.17 ± 0.03 to 0.61 ± 0.07 μmol/g FW) (Figure 1B), while in white cabbages purchased from ORG_1_ (Figure 1C), Allyl did not significantly change over time. In red cabbage, no significant changes were observable for Allyl (Figure 1D–F). 3MSOP as the other main GLS of white cabbage stayed the same over the sampling period in cabbages from CON_2_ and ORG_1_, but displayed increased levels in S4 and S5 of cabbages from CON_1_ compared to the S1-S3 samples (Figure 1A). In red cabbage, 3MSOP did not change across the consecutive sampling dates (Figure 1D–F). 4MSOB as a major GLS in red cabbage displayed reduced content in S2-S4 compared to S1 in CON_1_-cabbage (Figure 1D), but did not change in cabbages from CON_2_ and ORG_1_. Likewise, in white cabbage of ORG_1_, 4MSOB was not affected. In white cabbage from CON_1_, 4MSOB slightly decreased from S1 to S5 but was highest in S6 (Figure 1A), and in CON_2_-cabbages, this GLS generally displayed similar levels over the sampling dates (Figure 1B). Similarly, 2OH3But varied significantly only in white and red cabbage procured from CON_1_ (Figure 1A,D), but not in cabbages from the other supermarkets. In CON_1_-cabbages, 2OH3But was highest in S1 (red cabbage) or S2 (white cabbage) samples, then decreased until S4 (red cabbage) or S5 (white cabbage) and then again increased in the last samples (by tendency in red cabbage; significantly in white cabbage). The indole GLS I3M generally had similar levels over the different sampling dates in white and red cabbages, as well (Figure 1).

### 3.2. Glucosinolate Hydrolysis Product Formation in White and Red Cabbages from Local Food Retailers

Resulting from the homogenization of the fresh cabbage material, GLSs in cabbages from the different food retailers were degraded. The most pronounced GLS hydrolysis products released from white and red cabbages included the ITCs (or follow-up products from ITC) 3-(methylsulfinyl)propyl ITC (3MSOP-ITC), 4-(methylsulfinyl)butyl ITC (4MSOB-ITC), 3-butenyl ITC (3But-ITC), and 5-vinyloxazolidine-2-thione (OZT), the nitriles 5-(methylsulfinyl)pentanenitrile (4MSOB-CN) and 4-(methylsulfinyl)butanenitrile (3MSOP-CN) and the ETNs 1-cyano-2,3-epthiopropane (CETP), 1-cyano-3,4-epithiobutane (CETB), and isomers of 1-cyano-2-hydroxy-3,4-epithiobutane (CHETB A, CHETB B) (Appendix A). Overall, the main GLS hydrolysis products in white cabbages were 3-MSOP-CN and 3MSOP-ITC, which were formed from 3MSOP and CETP, originating from the GLS Allyl (Figure 2A–C).

Red cabbages released mainly 4MSOB-CN and 4MSOB-ITC, originating from 4MSOB and CETP, but also 3MSOP-products and CETB and CHETB were often released in higher amounts (Figure 2D–F). Usually, homogenized red cabbages released more GLS-hydrolysis products compared to white cabbages (Figure 2 and Figure 3). The formation of the cancer-preventive ITC 4MSOB-ITC was highest in red cabbages procured from CON_1_ in the S1 sample (1.06 ± 0.25 μmol/g FW), where it was also the main GLS-hydrolysis product. Although still being the main GLS-hydrolysis product in some samples, less 4MSOB-ITC was released in CON_2_- and ORG_1_- cabbages (up to 0.33 ± 0.18 μmol/g FW in S3 from CON_2_ and up to 0.44 ± 0.16 μmol/g FW in S2 from ORG_1_) (Figure 2D–F). The formation of GLS hydrolysis products in cabbages purchased from the three different supermarkets generally varied over the course of the sampling period (Figure 2 and Figure 3). Overall, the S5 sample of red cabbages with 2.68 ± 0.57 µmol/g FW displayed the highest total level of released GLS-hydrolysis products.

Regarding white cabbages, total ITC concentrations were highest with up to 0.50 ± 0.19 µmol/g FW in the later samples S4 and S5 from CON_1_, where they were also the main GLS hydrolysis product type. In addition, total nitrile levels were higher in these later samples. Total ETN levels were also highest in S5 (Figure 3A). In white cabbages from CON_2_, total ITC levels did not change during the sampling season, while ETN levels increased in later samples (S3–S6) and nitriles only displayed increased levels in S4 (Figure 3B). In organic white cabbages from ORG_1_, cabbages from the first sampling date (S1) released with up to 0.96 ± 0.14 µmol/g FW mainly ITCs, which was generally the highest observed ITC level in white cabbages. On the other hand, samples S3, S5, and S6 only showed low levels of ITC formation (0.02–0.06 µmol/g FW) with nitriles and ETNs as the most dominant GLS-hydrolysis products (Figure 3C).

Red cabbages from the first samples (S1–S3) usually mainly released ITCs, while later samples (S3–S6) mainly released nitriles or ETNs (Figure 3D–F). More specifically, the total ITC formation was highest in the first samples and peaked in cabbages from S1 (CON_1_ and CON_2_) or S2 (ORG_1_), while later samples released much lower total ITCs (Figure 3D–E). In red cabbages from CON_1_, ETNs and nitriles displayed increased levels in S5 and S6 cabbages compared to S1–S4 (Figure 3D). Comparably, in CON_2_, the red cabbages total ETN formation was also higher in S5, but nitriles were not affected (Figure 3E). In organic red cabbages from ORG_1_, total nitriles were also highest in S5, while ETN-formation fluctuated and displayed increased levels in S2 and S5 compared to S3 and S6 (Figure 3F).

Regarding the ratios (%) of total EPTs, nitriles, and ITCs relative to the total amount of formed GLS hydrolysis products, usually the relative ITC formation was higher in the first samples of cabbages, where they were often the main GLS hydrolysis products, but ITC formation decreased towards later sampling dates. Relative nitrile formation often behaved the other way around and was higher in later samples compared to early samples (Figure 4). More specifically, in white cabbages from CON_1_, the % of ITCs more than halved, while relative nitrile levels more than tripled from S1 to S6 and relative ETN formation was slightly increased in S2 compared to the other samples (Figure 4A). In white cabbages from CON_2_, the relative ITC formation was highest in S2 with 63 ± 11% and then decreased to S5 by 56% to 28 ± 9% of ITC formation. While relative nitrile formation was hardly affected, relative ETN formation increased from S1 to S5 to 46 ± 10% of ETN formation in S5 (Figure 4B). In organic white cabbages from ORG_1_, GLSs also mainly released ITCs with 75 ± 3% in S1, while cabbages from later samples (S3, S5, and S6) mainly released nitriles with up to 65 ± 7% (in S5) (Figure 4C). Red cabbages showed a very similar GLS hydrolysis behavior: With up to 75 ± 7% of ITC formation (S1 from CON_1_) the first samples (S1-S3) released mainly ITCs and the formation decreased towards later sampling dates, while nitrile formation increased in reverse with later samples to up to 64 ± 8% (S5 from ORG_1_).

The relative ETN release was not affected in red cabbages from CON_2_ and ORG_1_, but increased with later samples in CON_1_ up to 45 ± 9% of all GLS products (in S5) (Figure 4D–F). As an indicator of ESP-activity, the relative formation of CETP, Allyl-ITC, and Allyl-CN were monitored, as well. In red cabbages from CON_1_ and ORG_1_, the relative release of CEPT increased from first to last samples, while in white cabbages, an increase from S1 to S2–S5 was found (Appendix A). The relative CETP formation was unaffected in red cabbages from CON_2_ and conventional white cabbages.

### 3.3. Glucosinolates and Glucosinolate Hydrolysis Products Formation in Freshly Harvested White and Red Cabbages

The GLS profile of freshly harvested white and red cabbages was similar to the commercial ones, and 3MSOP and Allyl were most dominant in white cabbage (Figure 5A), whilst 2OH3But and 4MSOB contributed the most towards the total GLS content of freshly harvested red cabbage (Figure 5D). The total GLS and GLS-hydrolysis product level of red cabbages was higher compared to the freshly harvested white cabbages. With regard to individual GLS hydrolysis product formation, the main GLS hydrolysis products released from homogenized freshly harvested white cabbages were CETP and 3MSOP-CN (Figure 5B) and from freshly harvested red cabbages 4MSOB-CN and 4MSOB-ITC (Figure 5E). Of the detected GLS hydrolysis products from freshly harvested white cabbages, 36 ± 13% were nitriles (mainly 3MSOP-CN), 34 ± 7% were ETNs, and 30 ± 9% were ITCs (Figure 5C), while in red cabbages GLSs were degraded to 52 ± 3% nitriles (mainly 4MSOB-CN), 28 ± 2% ETNs, and 20 ± 4% ITCs (Figure 5F).

## 4. Discussion

In this study, the GLS content and formation of GLS hydrolysis products was evaluated in commercial white and red cabbages purchased from two conventional and one organic supermarket in Germany over a period of 3 months and compared to freshly harvested cabbages. In general, the composition of individual GLSs in red and white cabbages among different food retailers displayed only slight fluctuations and the GLS profile and levels were similar over the six sampling periods (Figure 1). With Allyl and 3MSOP being the main GLS in commercial and freshly harvested white cabbages and red cabbages being rich in 4MSOB and 2OH3But, (but also of Allyl, 3But, 3MSOP, and I3M), the GLS profiles and levels were in accordance to previous reports [14,24]. The small variability in GLS levels is an unexpected observation, as the analyzed white and red cabbages differed in genotype, came from different regions, were cultivated on different soil types using different fertilizers and storage practices, and were also purchased from different food retailers that belonged to different food trading companies (Table 2, Appendix A). Previous studies showed that GLS levels in *Brassica oleracea* vegetables are affected by cultivar (genotype) [14,25], nutrient supply [26,27], climatic conditions [17,20,28], as well as storage conditions [29,30]. As the variability of the GLSs was relatively low, it is suspected that genotypes were similar in their initial GLS concentrations and that also cultivation practices and storage conditions had no major effect on the GLS content of the cabbages, when they were finally sold in the supermarket. On the other hand, long-term storage (2 °C, 95% of relative humidity, up to 100 days) was shown to decrease the GLS content in Chinese cabbage (*Brassica rap*a L. spp. *pekinensis*), with GLSs being more stable in cabbages stored under a controlled atmosphere (CA) (2% O_2_ and 2% CO_2_) [31]. Accordingly, Osher et al. (2018) even reported an increase for aliphatic ITC-formation in cabbage (*B. oleracea*) stored at 1 °C under CA for 60 days (CA: 2% O_2_, 5% CO_2_), while ITC formation declined when stored under normal atmosphere (up to 45 and 72% decline in Allyl-ITC after 60 and 90 days, respectively) [32]. In the present study, I3M levels of organic white and red cabbages from ORG_1_ were often higher, compared to the cabbages from the conventional food retailers CON_1_ and CON_2_ (Figure 1C). Likewise, using NMR spectroscopy, Lucarini et al. (2020) recently also found nearly 3 times as much I3M in organic broccoli compared to conventionally grown ones [33]. In that study, the main difference in both farming practices was the fertilization practice, as no pesticides were used. While the same amount of nitrogen was supplied to the soil, for conventional broccoli with 0.2 t/ha urea and 15 t/ha bovine manure were applied, while organic grown broccoli was fertilized with 28 t/ha [33]. Nevertheless, increased I3M biosynthesis could be also explained due to the absence of chemical pesticides in organic cultivation practices, resulting in the stimulation of indole GLS biosynthesis upon herbivory damage via the methyl jasmonate signalling pathway [34].

Upon homogenization, GLSs were hydrolyzed in cabbages, yielding nitriles, ITC (or breakdown product thereof), and ETNs. Usually, the recovery of aliphatic GLS hydrolysis products was good with recoveries in a range of 60–130%. However, in some samples, low recoveries of aliphatic GLS hydrolysis samples were also observed (for example, S3 and S6 white cabbage samples and the S3 red cabbage sample from ORG_1_) (Figure 1 and Figure 2). Regarding this observation, myrosinase activity in these samples was probably low, therefore, resulting in a low recovery of hydrolysis products. In pre-experiments performed for the current study, the recovery of GLS hydrolysis products did not benefit from longer incubation times. Probably due to chemical instabilities of the products [35,36], it is likely that the myrosinase activity decreases with incubation time and that the initial myrosinase activity might be a major factor for the recovery of products. This hypothesis is further supported by the observation that sulfate, which is released during GLS hydrolysis, is a competitive inhibitor of myrosinase activity [37].

With regard to the differences in GLS hydrolysis products in the different samples, in contrast to GLS, the formation of GLS hydrolysis products varied strongly between the purchase dates (S1–S6; Figure 2). Especially during the first samples (S1–S3) mainly ITCs were released from red cabbages, while in later samples nitriles were preferentially formed (S5, S6) (Figure 3). Likewise, the relative ITC formation in white and red cabbages generally decreased until the last sample, while nitriles were usually the major hydrolysis products (Figure 4). These results show that in contrast to some previous reports [14,38], ITCs can be the main hydrolysis products in cabbages, as this was the case for cabbages purchased in early autumn. Here, red cabbage could be an excellent source for cancer-preventive 4MSOB-ITC (sulforaphane) (up to 1.06 μmol/g FW in red cabbage from CON_1_ at S1), releasing levels, which were 6-times higher compared to mature broccoli and comparable to the 4MSOB-ITC release from broccoli sprouts [14].

With regard to the high nitrile and ETN formation at later purchase dates, the ESP protein activity is made responsible for ETN-release from alkenyl GLS and for increased formation in simple nitriles from non-alkenyl GLS [39,40]. Therefore, it was suspected that the ESP-activity increased towards later samples, while it remained low in the first samples. When regarding the relative formation of Allyl hydrolysis products, as an indicator of ESP activity, it can be further supported that the ESP activity increased with later purchase dates in red cabbage from CON_1_ and organic cabbage, while it was not considerably affected in conventional white cabbage (Appendix A). As the relative nitrile formation significantly increased in white cabbage from CON_1_ in later samples (Figure 4A) (but relative CETP release not, Appendix A), it is suspected, that next to the ESP activity, also other factors influence GLS hydrolysis, which could have resulted in changes in hydrolysis product behavior from S1 to S6 due to their variation. As especially nitriles increased (Figure 4A), it is suspected that white cabbage contains nitrile specifier proteins, which are involved in nitrile formation in *Arabidopsis thaliana* [41]. This suspicion is strengthened by the observation that *Brassica oleracea* contains a gene with a homology of 80% compared to the nitrile specifier protein 1 of *A. thaliana* [42]. Further, this hypothesis is supported by a recent study, which could neither prove the nitrile specifier protein activity for three *B. oleracea* ESP isoforms in vitro nor in vivo, but nitrile formation from alkyl GLS was observed [43].

To date, there is little data how pre- and postharvest factors affect glucosinolate hydrolysis. Freshly harvested white and red cabbages showed a similar GLS hydrolysis behavior compared to the supermarket cabbages purchased during similar dates (S5 white cabbage, S4 red cabbage) (Figure 5; Figure 2, Figure 3 and Figure 4). Due to this finding and due to the different storage conditions of growers and retailers, in the present study storage does not seem to be the factor that caused a reduced ITC release in cabbages purchased in later autumn. With regard to preharvest factors, nitrogen and sulfur supply affected the release of GLS hydrolysis products in the ETN-producer Chinese cabbage (*Brassica rapa* L. ssp. *pekinensis*) and ITCs were reduced in response to the increasing N and decreasing S supply [44]. In pak choi (*B. rapa* subsp. *chinensis* (L.) Hanelt), which also mainly released ETN, the ITC/CN and ITC/ETN ratio increased with the increasing sulfur supply [45]. With regard to the present study, it is unlikely that differences in fertilization were responsible for the observed changes in the GLS hydrolysis behavior as also the red cabbage from CON_2_ which originated from the same grower (Table 2, Appendix A) showed reduced % ITC release at later purchase dates (Figure 4E). Moreover, herbivore feeding can affect the GLS hydrolysis behavior, and simple nitrile formation was shown to increase in response to the specialist insect feeding (*Pieris rapae*) in *Arabidopsis thaliana* Col-0 [46], while ITC-emitting plants appear to be better defended against generalist herbivores [47]. However, as organic and conventional cabbages displayed a similar hydrolysis behavior (Figure 4D–F), it is suspected that climatic conditions such as reduced radiation or decreasing temperatures across the autumn season might be responsible for the observed shifts. Moreover, all of the conventional cabbages originated from the Dithmarschen region (Schleswig-Holstein, Germany; Appendix A), which is characterized by a coastal climate and marsh soil (being ideal conditions for growing cabbages). It is the largest coherent cabbage-growing area in Europe. In Appendix A, the climatic data of the presumable growing season in 2019 is presented. Consequently, it is likely that temperature and radiation interact with regard to the GLS hydrolysis behavior. Recently, Jasper et al. (2020) showed that at higher temperatures during growth, more GLS hydrolysis products were formed from rocket (*Eruca sativa*), while GLS levels were less affected [48]. Likewise, Ku et al. (2013) reported different ITC conversion rates in broccoli grown in 2 different years and linked this to different climatic conditions. Unfortunately, in that study, nitriles were not analyzed and therefore, conversion rates could have been also affected by changes of the myrosinase activity [49]. As organic cabbages which originated from different regions in Germany (Brandenburg, Mecklenburg-Western Pomerania, Schleswig-Holstein; Table 2 and Appendix A) also showed similar changes in the GLS hydrolysis behavior compared to the conventional cabbages originating from the Dithmarschen region, it can be assumed that the results obtained in this study might be also valid for other countries and regions with similar climatic conditions. The specific role of climatic growth conditions on GLS hydrolysis in *B. oleracea* vegetables will need to be evaluated in future studies.

## 5. Conclusions

Current findings in this study have highlighted the great diversity, particularly of the GLS hydrolysis behavior in white and red cabbages between the different supermarkets over the six sampling periods: Whilst the GLS composition and content remained similar between the different food retailers, the composition and content of the individual hydrolysis products formed varied across the season and high ITC levels were generally noted in early sampling periods (early September) and decreased, particularly in red cabbages over time. Here, the increased specifier protein activity is made responsible for the reduced ITC-release.

In conclusion, with regard to their potential to release more ITCs, consumption of commercial cabbages purchased in early autumn could be healthier options than those purchased in later autumn months. The fact that ITCs can be preferentially formed in earlier autumn months, but hardly towards the end of autumn, underlines the need to unravel the factors that affect the GLS-hydrolysis outcome. The results of this study might also help growers and food companies produce cabbages and products with more pungency due to a higher ITC formation. Due to the potential of red cabbage to form high rates of health-promoting 4MSOB-ITC in cabbages purchased in early autumn, red cabbage consumption could be an alternative for people who dislike broccoli.

## Figures and Tables

**Figure 1 foods-09-01682-f001:**
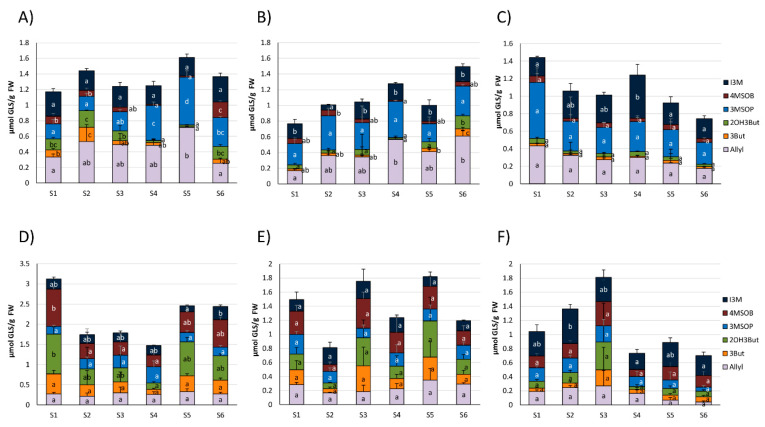
Glucosinolate (GLS) profile in white (**A**–**C**) and red cabbages (**D**–**F**) from conventional (**A**,**B**,**D**,**E**) and organic supermarkets (**C**,**F**) on different sampling dates (S1–S6). Exact sampling dates can be found in Table 1. A) and D) Represent conventional supermarket 1 (CON_1_), B) and E) stand for conventional supermarket 2 (CON_2_), and C) and F) show results from organic supermarket 1 (ORG_1_). Each color in the bar of the given bar chart represents the mean plus standard deviation (SD) of the GLSs from three cabbage heads from the same supermarket (*n* = 3). Lower case letters indicate significant differences in means between the levels of a GLS on different sampling dates, as tested by ANOVA and Tukey HSD test at the *p* ≤ 0.05 level. Abbreviations: FW: Fresh weight; abbreviations of compounds as listed in Table 3.

**Figure 2 foods-09-01682-f002:**
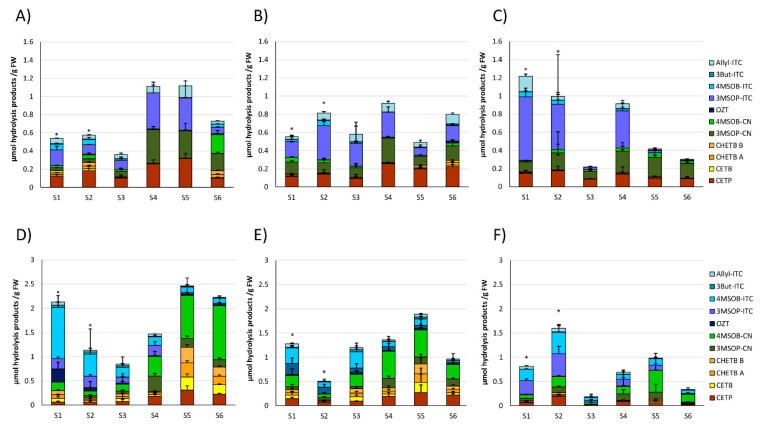
Main glucosinolate (GLS) hydrolysis products released from white (**A**–**C**) and red cabbages (**D**–**F**) from conventional (**A**,**B**,**D**,**E**) and organic supermarkets (**C**,**F**) on different sampling dates (S1–S6). Exact sampling dates can be found in Table 1. (**A**,**D**) Represent conventional supermarket 1 (CON_1_), (**B**,**E**) stand for conventional supermarket 2 (CON_2_), and (**C**,**F**) show results from organic supermarket 1 (ORG_1_). Each color in the bar of the given bar chart represents the mean plus standard deviation (SD) of the GLS hydrolysis products from three cabbage heads from the same supermarket (*n* = 3). Abbreviations: FW: Fresh weight; abbreviations of compounds are listed in Table 3. * Samples were analyzed from 1 g of sample homogenate (instead of 0.5 g).

**Figure 3 foods-09-01682-f003:**
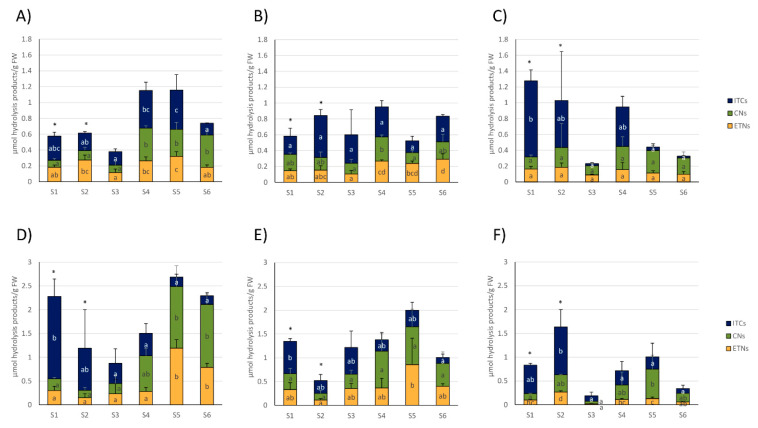
Formation of total isothiocyanates (ITCs), nitriles (CNs), and epithionitriles (ETNs) from white (**A**–**C**) and red cabbages (**D**–**F**) from conventional (**A**,**B**,**D**,**E**) and organic supermarkets (**C**,**F**) on different sampling dates (S1–S6). Exact sampling dates can be found in Table 1. (**A**,**D**) Represent conventional supermarket 1 (CON_1_), (**B**,**E**) stand for conventional supermarket 2 (CON_2_), and (**C**,**F**) show results from organic supermarket 1 (ORG_1_). Each color in the bar of the given bar chart represents the mean minus standard deviation (SD) of the total ITC, CN, or ETN levels from three cabbage heads from the same supermarket (*n* = 3). Different lower case letters indicate significant differences in means between total ITCs, CNs, and ETNs levels in different samples, as tested by ANOVA and Tukey’s HSD test at the *p* ≤ 0.05 level. FW: Fresh weight. * Samples were analyzed from 1 g of sample homogenate (instead of 0.5 g).

**Figure 4 foods-09-01682-f004:**
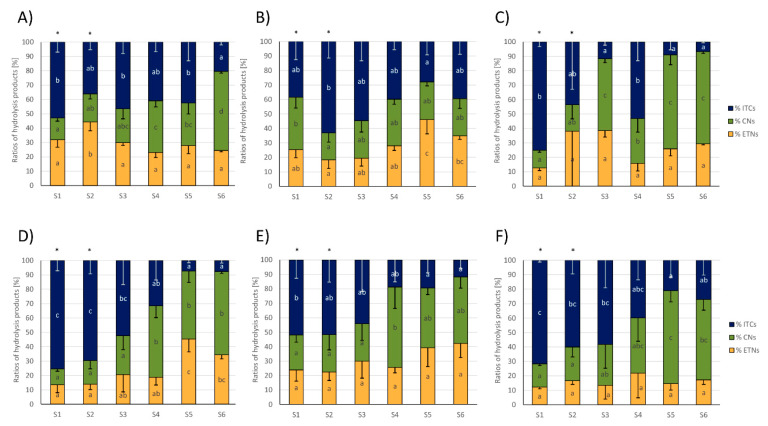
Total glucosinolate (GLS) hydrolysis product ratios (percentage of total isothiocyanates (ITCs), nitriles (CNs), and epithionitriles (ETNs)) from white (**A**–**C**) and red cabbages (**D**–**F**) from conventional (**A**,**B**,**D**,**E**) and organic supermarkets (**C**,**F**) on different sampling dates (S1–S6). Exact sampling dates can be found in Table 1. (**A**,**D**) Represent conventional supermarket 1 (CON_1_), (**B**,**E**) stand for conventional supermarket 2 (CON_2_), and (**C**,**F**) show results from organic supermarket 1 (ORG_1_). Each color in the bar of the given bar chart represents the mean minus standard deviation (SD) of the GLS hydrolysis products from three cabbage heads from the same supermarket (*n* = 3). Different lower case letters indicate significant differences in means between the ratios of total ITCs, CNs, and ETNs in different samples, as tested by ANOVA and Tukey’s HSD test at the *p* ≤ 0.05 level. FW: Fresh weight. * Samples were analyzed from 1 g of sample homogenate (instead of 0.5 g).

**Figure 5 foods-09-01682-f005:**
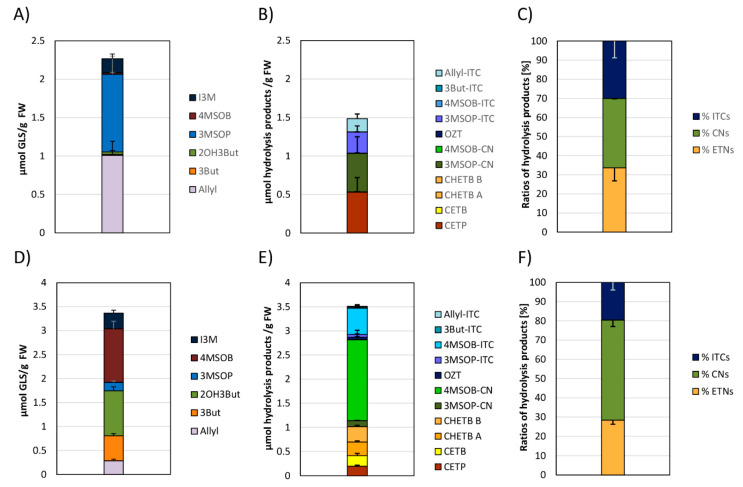
Glucosinolate (GLS) (**A**,**D**) and their absolute (**B**,**E**) and relative (**C**,**F**) hydrolysis product formation in white (**A**–**C**) and red cabbages (**D**–**F**) harvested freshly from the field. Each color in the bar of the given bar chart represents the mean plus standard deviation (SD) of the GLS (**A**,**D**), their respective hydrolysis products (**B**,**E**), or the ratio of relative isothiocyanates (ITCs), nitriles (CNs), and epithionitriles (ETNs) (**C**,**F**) from three cabbage heads freshly harvested from the field (*n* = 3). Abbreviations: FW: Fresh weight; abbreviations of compounds are listed in Table 3.

**Table 1 foods-09-01682-t001:** Overview of the dates of the purchased and harvested red and white cabbage heads.

		Date of Purchase	
Supplier	Abbreviation	White Cabbage	Red Cabbage
CON_1_	S1	04.09.2019	04.09.2019
	S2	16.09.2019	16.09.2019
	S3	30.09.2019	30.09.2019
	S4	21.10.2019	21.10.2019
	S5	04.11.2019	04.11.2019
	S6	18.11.2019	18.11.2019
CON_2_	S1	05.09.2019	05.09.2019
	S2	16.09.2019	16.09.2019
	S3	30.09.2019	30.09.2019/01.10.2019
	S4	21.10.2019	21.10.2019/23.10.2019
	S5	04.11.2019	4.11.2019/06.11.2019
	S6	18.11.2019	18.11.2019
ORG_1_	S1	09.09.2019	09.09.2019
	S2	19.09.2019	19.09.2019
	S3	01.10.2019	01.10.2019
	S4	23.10.2019	23.10.2019
	S5	06.11.2019	06.11.2019
	S6	20.11.2019	20.11.2019
**Fresh harvest from field (IGZ)**		**Date of harvest**	
		**White Cabbage**	**Red Cabbage**
		29.10.2019	10.10.2019

**Table 2 foods-09-01682-t002:** Total overview of the German origin, cultivation, and storage conditions of commercial red and white cabbages and red and white cabbages from the field experiment. Purchase dates of S1–S6 can be found in Table 1.

Supplier	Origin of Cultivation	Samples/Date of Harvest (IGZ)	Harvest Season	Cabbage Type	Cabbage Genotype	Soil Type and Field	Fertilizer	Certifica-Tion Mark	Storage Conditions *^1^	Storage Conditions *^2^
**CON_1_**	25792, Neuenkirchen Schleswig-Holstein	S1–S3	Summer, autumn	(1), (2)	White cabbage hybrids	Sea marsh	Urea, CAN, phosphate and potash	QS	(i)	20 °C day/night
		S1–S4			Red cabbage hybrids					
**CON_1_**	25709, Helse Schleswig-Holstein	S4–S6	Autumn, winter	(2), (3)	White cabbage hybrids	Sea marsh, pH-value 7.0–7.4	PKS fertilizer (blends), ammonium nitrate and urea, calcium cyanamide	QS	(ii)	20 °C day/night
		S5, S6			Red cabbage hybrids					
**CON_2_**	25792, Neuenkirchen Schleswig-Holstein	S1–S6	Summer, autumn, winter	(1), (2), (3)	Red and white cabbage hybrids	Sea marsh	Urea, CAN, phosphate and potash	QS	(iii)	7–10 °C day/night in cooling counter, max. 1 week
**ORG_1_**	17237, Blankensee Mecklenburg-Western Pomerania	S1, S4, S5	Autumn	(2)	White cabbage hybrids	Sandy loam	Hair-meal pellets	Bioland	(iv)	10 °C day/night in cooling counter, max. 1 week
		S1			Red cabbage hybrids					
**ORG_1_**	25761, Hedwigenkoog Schleswig-Holstein	S2, S3	Autumn	(2)	White cabbage, non-hybrid	Sea Marsh	Compost	Demeter	(iv)	Day/night in cooling counter, max. 1 week
		S2, S3, S4, S5			Red cabbage, non-hybrid					
**ORG_1_**	15306, Vierlinden Brandenburg	S6	Autumn	(2)	White cabbage hybrids	Sandy loam	Hair-meal pellets	Bioland	(v)	Day/night in cooling counter, max. 1 week
**ORG_1_**	14715, Seeblick Brandenburg	S6	Autumn	(2)	Red cabbage, non-hybrids	Sandy loam	Compost	Demeter	(iv)	Day/night in cooling counter, max. 1 week
**IGZ**	14979, Großbeeren Brandenburg	10 October 2019	Autumn	(2)	Red cabbage (Redma RZ F1), hybrid	Silty loam	CAN, patentkali	-	No storage	No storage
**IGZ**	14979, Großbeeren Brandenburg	29 October 2019	Autumn	(2)	White cabbage (Dottenfelder Dauer), non-hybrid	Silty loam	Vinasse	-	No storage	No storage

*^1^ Before selling, *^2^ during selling; CAN-calcium ammonium nitrate. (i) 6–8 °C for 1–2 days after harvest, storage for 1–2 days (intermediate trade), then storage at 6–8 °C (cold storage warehouse). (1) Early cabbage (fresh harvest). (ii) 6–7 °C for 1–2 days in central warehouse. (2) Late cabbage (fresh harvest). (iii) 6–8 °C for 1–2 days after harvest, then storage at 6–8 °C in cold storage warehouse. (3) Long-term stored (stored). (iv) 2–6 °C for 1–2 days after harvest, then storage at 4–7 °C for 1–3 days in cold storage warehouse (max 48 h). (v) 0.1–0.3 °C cold storage warehouse.

**Table 3 foods-09-01682-t003:** Structures of main cabbage glucosinolates, their abbreviations, and their hydrolysis products analyzed in the present study. N.d.: Not detected.

	Glucosinolates (GLSs)	Corresponding Breakdown Products
		Isothiocyanate (ITC)	Nitrile	Epithionitrile (ETN)
Structure	Abbreviation	Name (trivial name)	Abbreviation	Name	Abbreviation	Name	Abbreviation	Name
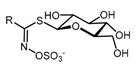								
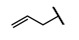	Allyl	allyl GLS (sinigrin)	Allyl-ITC	2-propenyl ITC	Allyl-CN	3-butenenitrile	CETP	1-cyano-2,3-epithiopropane
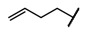	3But	3-butenyl GLS (gluconapin)	3But-ITC	3-butenyl ITC	3But-CN	4-pentenenitrile	CETB	1-cyano-3,4-epithiobutane
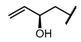	2OH3But	2-(*R*)-2-hydroxy-3-butenyl GLS (progoitrin)	OZT	5-vinyl-1,3-oxazolidine-2-thione		3-hydroxy-pentenenitrile	CHETB ACHETB B	1-cyano-2-hydroxy-3,4-epithiobutane
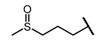	3MSOP	3-(methylsulfinyl)propyl GLS (glucoiberin)	3MSOP-ITC	3-(methylsulfinyl)-propyl ITC	3MSOP-CN	4-(methylsulfinyl)-butanenitrile	-	-
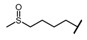	4MSOB	4-(methylsulfinyl)butyl GLS (glucoraphanin)	4MSOB-ITC	4-(methylsulfinyl)butyl ITC	4MSOB-CN	5-(methylsulfinyl)-pentanenitrile	-	-
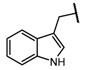	I3M	indol-3-ylmethyl GLS (glucobrassicin)		n.d.		indole-3-acetonitrile	-	-

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
