# Peer review of "Seasonal Variation of Glucosinolate Hydrolysis Products in Commercial White and Red Cabbages (Brassica oleracea var. capitata)"

_foods, 2020, doi:10.3390/foods9111682_

Round 1
Reviewer 1 Report
Dear authors
I would like to see the following information concerning your research paper:
- the correlation of the harvest day of each cabbage with GLS and its hydrolysates
- the correlation of each post harvest day of each cabbage with its GLS content till purchase day (consumption day is another aspesct)
- fresh organic cabbage specimens experimentation that can be correlated with all other controls and samples
- a table(s) of the measurements obtained from your experiments before the figures
- More extended discussion on your results, if applicable, since these findings have been obtained in Germany. Climate conditions differ from country to country, is there a correlation on your results and the climate? Are your findinds applicable to more areas than Germany?
Author Response
Dear authors
I would like to see the following information concerning your research paper:
Comment 1:
- the correlation of the harvest day of each cabbage with GLS and its hydrolysates
Reply: First of all we would like to thank the reviewer for his/her valuable time to review our manuscript and helping us to improve the manuscript! In the present study, cabbages were taken as samples from supermarkets and analyzed regarding glucosinolate hydrolysis behavior. Information on the origin/storage after harvest and before selling etc. was obtained by interrogating service staff at the supermarkets and by contacting growers. Thus, the information on the cabbages presented in Table S1 was compiled. Unfortunately, as the exact harvest day of the commercial cabbages is not known, only a correlation between the purchase date was made (as presented in Table 1 and Figures 1-4). However, from Table S1 (column Q, and R) it can be assumed that the range between harvest and purchase date usually was similar for the first five sampling dates (S1-S5) (ranging usually from 4-8 days). The last sampling (S6) was stored cabbage (which was likely stored about a month according to typical procedures in German retailing).
Comment 2:
- the correlation of each post harvest day of each cabbage with its GLS content till purchase day (consumption day is another aspect)
Reply: Again, the information on the storage duration between harvest and purchase day (column Q, and R in Table S1) has often a high range and an uncertainty (for example could be between 3-7 days as for organic cabbages at the first sampling). Consequently, this data is not suitable for correlations between storage duration and glucosinolate content and hydrolysis. However, as glucosinolate content and hydrolysis behavior is similar between S5 and S6, it can be assumed that cool storage had no dramatic effect on glucosinolate hydrolysis. This will be investigated in more detail in a future project.
Comment 3:
- fresh organic cabbage specimens experimentation that can be correlated with all other controls and samples
Reply: We are very sorry, but we only analyzed fresh conventionally grown cabbage (Figure 5), as organic cultivation according to EU legislation is not possible at the research site IGZ at the moment. In a future experiment on storage we will consider organic cabbage as well.
Comment 4:
- a table(s) of the measurements obtained from your experiments before the figures
Reply: We have now prepared a Supplemental Table giving the respective glucosinolate data (Table S2) and the data of the hydrolysis products (Table S3).
Comment 5:
- More extended discussion on your results, if applicable, since these findings have been obtained in Germany. Climate conditions differ from country to country, is there a correlation on your results and the climate? Are your findings applicable to more areas than Germany?
Reply: We thank the reviewer for the comment. Indeed, we hypothesized that results will correlate with the climatic conditions. All of the conventional cabbages and many of the samples of the organic cabbage came from the same region in Germany, the Dithmarschen region in North-Western Germany (Table S1). About one third of all cabbages produced in Germany are grown in this region, making it one of the most important regions for cabbage production. This is due to the climatic conditions, mainly because of a coastal climate with regular rainfall and marsh soil, which is ideal for growing cabbages. Therefore, it can be assumed that our results also apply for other regions with similar climatic conditions, however, there is no proof. In the Table S4, we now give climatic data from summer and autumn 2019 from Elpersbüttel, which is located in the Dithmarschen region. In the discussion we included some discussion about this (L. 499-504 and L. 509-514).
Reviewer 2 Report
The manuscript “Seasonal Variation of Glucosinolate Hydrolysis Products in Brassica oleracea var. capitata from Regional Food Retailers in Germany” describes the glucosinolate and glucosinolate hydrolysis products content in red and white cabbages from three different food retailers monitored from September to November.
Although the manuscript is clear and well described and organized, the novelty and significance of content is not high.
The analysis is limited to Germany products and focused only on glucosinolates and their hydrolysis products among the many having potential beneficial properties.
The authors are strongly encouraged to address the following comment in improving the manuscript:
- Describe on which criteria they select the three supermarkets given the fact that results can suffer to sample selection bias.
- Provide more discussion on how results can be useful to companies and retailers in add value to the product (e.g., product differentiation)
- Results discussion, it is too speculative. More references and scientific soundness is needed.
Author Response
Comment 1: The manuscript “Seasonal Variation of Glucosinolate Hydrolysis Products in Brassica oleracea var. capitata from Regional Food Retailers in Germany” describes the glucosinolate and glucosinolate hydrolysis products content in red and white cabbages from three different food retailers monitored from September to November.
Although the manuscript is clear and well described and organized, the novelty and significance of content is not high.
The analysis is limited to Germany products and focused only on glucosinolates and their hydrolysis products among the many having potential beneficial properties.
Reply: We thank the reviewer for his/her valuable time to review our manuscript. To the best of knowledge, our study is the first study that reveals that the glucosinolate hydrolysis behavior in commercial cabbages can change from isothiocyanate formation to epithionitrile/nitrile formation due to the season. The study reveals that cabbages in early autumn released more isothiocyanates than in late autumn.. The study focused only on German cabbages. All of the conventional cabbages and many of the samplings of the organic cabbage came from the same region in Germany, the Dithmarschen region (Table S1). In this region, about one third of all cabbages produced in Germany are grown and it is the largest coherent cabbage-growing area in Europe. This is due to the climatic conditions, as it has a coastal climate with regular rainfall and marsh soil, which is ideal for growing cabbages. Therefore, it can be assumed that our results also apply for other regions with similar climatic conditions, but there is no proof. In the Table S4, we have listed/included climatic data from summer and autumn 2019 from Elpersbüttel, which is located in the Dithmarschen region. In the discussion we have included some discussion about this (L. 499-504 and L. 509-514).
Comment 2:
The authors are strongly encouraged to address the following comment in improving the manuscript:
- Describe on which criteria they select the three supermarkets given the fact that results can suffer to sample selection bias.
Reply: The supermarkets were selected with regard to different German trading companies. The two conventional supermarkets selected, belong to the two biggest food trading companies in Germany. The organic supermarket selected, also belongs to one of the biggest organic food supermarket chains in Germany. It was also selected due to logistic reasons. We have included this information in L 115-117.
Comment 3:
- Provide more discussion on how results can be useful to companies and retailers in add value to the product (e.g., product differentiation)
Reply: According to cabbage retailers, food trading companies are mainly interested in visual quality of cabbages at the moment, but not on nutritive quality. Because of the EU legislation, it is not allowed to make any health related claim or claims on phytochemical content (Health Claims Regulation (EC) No 1924/2006) unless these claims would be allowed by the EU after applying for it. Nevertheless, if someone would like to produce a more pungent” cabbage, then cultivation in summer might be the right way. Nevertheless, surveys would have to be considered, in order to extract information as to whether the consumer would prefer such a pungent product. In the conclusion we now write “The results of this study might help growers and food companies to produce cabbages and products with more pungency due to higher ITC formation.”
Comment 4:
- Results discussion, it is too speculative. More references and scientific soundness is needed.
Reply: The discussion was carefully revised, references were added were needed to make the discussion less speculative.
Reviewer 3 Report
Numerous studies have highlighted a significant inverse correlation between the consumption of vegetables from the crucifereae family (broccoli, cabbage, cauliflower, etc.) and the reduced risk of developing neoplastic and cardiovascular diseases. The positive effects are to be attributed to the presence of glucosinolates and derivatives contained in these plants. In particular, isothiocyanates are a group of substances deriving from glucosinolates with chemoprotective activity. In this study, the profiles of glucosinolates and their hydrolysis products in red and white cabbage were monitored during the autumn season.
The results obtained showed variations in the content of isothiocyanates during the season, suggesting valid indications on the most suitable period for their food consumption in order to benefit from their effects.
The manuscript is suitable for the Journal target.
The reported work plan is valid and provides interesting results.
The experimental part is presented correctly and reports the data analysis.
I have no comments on this work.
Author Response
We thank the reviewer for his/her valuable time to review our manuscript.
Round 2
Reviewer 1 Report
Comment made and explanations given are satisfactory.
Reviewer 2 Report
Authors partially satisfied my comments. Overall the manuscript is fine for pubblication.
This manuscript is a resubmission of an earlier submission. The following is a list of the peer review reports and author responses from that submission.
Round 1
Reviewer 1 Report
The manuscript “Seasonal Variation of Glucosinolate Hydrolysis Products in Brassica oleracea var. capitata from Regional Food Retailers in Germany” submitted to Foods describes the glucosinolate and glucosinolate hydrolysis products content in red and white cabbages from three different food retailers monitored from September to November. Although the manuscript is clear and well described, the novelty and significance of content is not high. Their analyses are limited to Germany products and focused only on glucosinolates and their hydrolysis products. Recently, in this journal it has been published a similar manuscript (Foods 2020, 9(10),1371; https://doi.org/10.3390/foods9101371), where the authors evaluated the content of several important health-promoting bioactive compounds. A wider evaluation of bioactive compounds in their materials should increase the significance of their results. The authors are strongly encouraged to evaluate this suggestion to improve their manuscript.
Beside these general comments I have some concerns. In particular:
Paragraph 2.6 statistical analysis, the authors should clarify the meaning of this sentence “All analyses were carried out in triplicate”. Do they mean technical or biological triplicate? Usually for metabolite analysis a higher number of biological replicates are required.
Figure 2, in the legend they mention that different lower case letters indicate significant differences in means between the levels of GLS hydrolysis products but I am unable to see the letters in the figure. Further, they should clarify the meaning of this sentence “*Samples were analyzed from 1 g of sample” present also in other figures. I deduced that the samples without *were analyzed using another quantity of sample. The authors should explain the reason why they did not standardize the procedure with all samples. Different quantity of sample can affect the analysis?
Discussion, it is too speculative.
Reviewer 2 Report
The authors have proposed a study about the composition of commercial cabbages (red and white) in terms of profitable compounds for the human health, such as isothiocyanates with regard to the period they are harvested.
The work has not novelty in the work as it can be seen that similar works have been done (Ref 17) more than 10 years ago. There are some plus information in this work but it is not highly remarkable. There a clear lack of recent references that could highlight the novelty of this topic.
The samples were taken from different supermarket what has increase the degree of variability in this study. Despite the deep research carried out to identify the source of the samples (Section 2.2), this is not considered a scientific approach. The study would have been more reliable if the samples would have been controlled directly from the harvesting in order to minimize the variability of the samples.
The discussion of the results is difficult to follow due to the high number of abbreviated parameters (it is recommended to include an abbreviation list), but also due to the length of the figure captions, etc; although the scientific explanations and the analyses that were made were deep and well conducted.
Other comments:
Section 2.4. Extend the explanation of this technique.
Results. The first paragraph together with Table 2 have to be included in Section 2.
Table S1 should be upload in a PDF in a summarized view.
Reviewer 3 Report
Glucosinolates are glucosidic compounds contained in Brassicaceae. Their enzymatic degradation gives rise to a complex mixture of compounds including isothiocyanates and their derivatives, substances that are still widely studied for the beneficial effects on health (antimicrobial, antidiabetic, chemopreventive properties) and for their possible inhibitory activity on tumor genesis and growth.
General comments
The experimental approach is not particularly original, however the results obtained are interesting from a food and health point of view. In fact, they show that ITCs are formed preferentially in the early autumn months, but hardly towards the end of autumn, suggesting the best period for their consumption.
Overall, the study is well structured and the results are supported by a detailed analysis of the data reported in the various figures.
However, the authors need to make some changes to the manuscript:
Title should be changed so as not to have only local interest, for example:
Seasonal Variation of Glucosinolate Hydrolysis Products in Commercial White and Red Cabbages Brassica oleracea var. capitata.
In the Materials and Methods section (2.1) specify what type the various chemicals listed are.
Reviewer 4 Report
The authors explore a timely topic quantifying/exploring the Glucosinolate profiles of several cabbage varieties across the seasons. The text is well written, research question clearly presented and manuscript deserve publication. However, I have some comments that would like authors address:
- Since such chemicals (Glucosinolate Hydrolysis Products) have health properties, I am wondering if the authors want to suggest some strategies to let consumers aware about this (e.g., labling), thus promoting the product (Brassica oleracea var. capitata) consumption;
- How logistic and distribution operators as well as retailers can benefit from authors' findings. Do authors have some suggestion about this?
- Lastly, authors may provide additional figure about the market size and value of cabbages products, as well as number of employees/operators involved in such market in order to point out the relevance on why investigate in this market.
Best Regards